# Design, 3D printing, and preclinical validation of an extraglottic ramp to facilitate blind orotracheal intubation in emergency airway management

Jorman H. Tejada-Perdomo [1,2,3*], Valentina Gutierrez-Perdomo[1,3], Juana V. Agudelo-Castro[1,3], Jorge A. Pérez-Gamboa[4], Alejandro Weinstein[5], Sebastián San Martín[6], Rodrigo Salas[7], Jorge A. Ramos-Castaneda[8,9]

**1** Department of Clinical Sciences, Universidad Surcolombiana, Neiva, Colombia, **2** Department of Anesthesiology, Hospital Universitario Hernando Moncaleano Perdomo, Neiva, Colombia, **3** Research Group Desarrollo Social, Salud Pública y Derechos Humanos, Universidad Surcolombiana, Neiva, Colombia, **4** Diseñador Industrial, Neiva, Colombia, **5** Department of Electronic Engineering, Universidad Técnica Federico Santa Maria, Valparaiso, Chile, **6** Center of Interdisciplinary Biomedical and Engineering Research for Health (MEDING), School of Medicine, Faculty of Medicine, Universidad de Valparaíso, Viña del Mar, Chile, **7** Center of Interdisciplinary Biomedical and Engineering Research for Health (MEDING), Biomedical Engineering School, Faculty of Engineering, Universidad de Valparaíso, Viña del Mar, Chile, **8** Research Group Innovación y Cuidado, Universidad Antonio Nariño, Neiva, Colombia, **9** Research Group Cuidar, Universidad Surcolombiana, Neiva, Colombia

* jorman.tejada@usco.edu.co

## Abstract

The use of devices that facilitate rapid airway isolation is essential when managing critical patients in emergencies. In recent years, additive manufacturing has emerged as an innovative, versatile, and accessible technology for developing medical devices. This study presents the design, development, and validation of an extraglottic medical device created using computer-aided design tools and stereolithographic 3D printing to facilitate blind intubation by first responders. The device was iteratively modeled and fabricated with biocompatible materials; validation in airway simulators and human cadaveric specimens assessed dimensions, friction, intubation technique, and learning curve, and ease of use was rated with a Likert scale. Ten iterations led to a final design with low friction and minimal cervical manipulation; ramp angle, cup geometry, and distal tip were optimized for tube passage, and BioMed Flex 80A showed high strength and anatomical compatibility. The final version is a safe, reusable, and functional alternative for airway management and blind orotracheal intubation, particularly in emergencies and resource-limited settings; clinical validation in live patients is still needed.

**Data availability statement:** All relevant data are within the paper.

**Funding:** JHTP received institutional support from Universidad Surcolombiana (USCO) (internal funding, no specific grant number). AW was partially funded by Agencia Nacional de Investigación y Desarrollo, Chile (ANID), BASAL grant AFB240002 to AW. The funders had no role in study design, data collection and analysis, decision to publish, or preparation of the manuscript. There was no additional external funding received for this study.

**Competing interests:** The authors have declared that no competing interests exist.

## Introduction

In recent years, additive manufacturing (AM), also known as three-dimensional (3D) printing, has seen growing development not only in engineering but also in medicine [1]. AM is a process that enables the fabrication of complex parts by adding material layer by layer based on 3D model data [2]. In the medical field, this manufacturing technique has enabled the creation of 3D models of tissues as well as patient-specific structures.

The first step in generating a prototype using AM is to create a three-dimensional model using computer-aided design (CAD) software. This design process aims to reproduce the physical 3D object with minimal material waste, thereby reducing production time and cost [3].

AM has been continuously refined and is gaining increasing prominence. Various techniques have been employed to design and create 3D-printed tissues or implants for medical applications [4], which requires a special focus on the use of materials that do not harm the patient, in other words, the use of biocompatible materials, as well as on bioprinting methods [5]. These processes are commonly referred to as biofabrication [5,6].

Currently, biofabrication enables the production of complex 3D tissues or structures with biological and mechanical properties, opening the door to countless potential applications in the medical field, such as the creation of implants, prostheses, anatomical models for educational purposes (e.g., airway trainers), and prototypes of medical devices [7].

On the other hand, orotracheal intubation involves placing a tube into the trachea for airway management. It is a fundamental pillar in the treatment of patients with severe trauma, most often performed using a laryngoscope [8].

There are well-established and widely accepted strategies for tracheal intubation, such as supraglottic devices (SGDs), videolaryngoscopes, and blind endotracheal intubation devices [9,10]. The success rate of blind intubation varies depending on the type of supraglottic device, the patient's characteristics, and the operator's skill [11–15]. In initial studies, which served as the starting point for this work, a medical device for blind orotracheal intubation, named the orolaryngeal ramp, was evaluated using simulation models, yielding satisfactory results [16,17].

This study aimed to describe the technological development process of an extraglottic medical device for blind intubation by creating a high-performance prototype using CAD methodology, selecting materials, employing rapid prototyping techniques based on additive manufacturing, and conducting validation tests in airway simulators and human cadavers.

## Materials and methods

We defined an a priori design objective—rapid (<30 s) and effective blind intubation with ≥95% success—guided by performance benchmarks reported for established supraglottic devices and time targets described in airway management literature [15,18,19]. A detailed analysis of the blind orotracheal intubation process was initially conducted, highlighting the challenges faced by non-expert personnel in airway

management, such as: the need to secure the airway quickly, the requirement for neck hyperextension maneuvers, fear of inserting devices into the oral cavity due to the risk of dental injury or bleeding-related trauma, and the early recognition of esophageal intubation [20,21].

The medical device consists of four parts: an external opening for inserting the orotracheal tube, a semi-rigid cup, an internal opening or laryngeal aperture, and a flap or epiglottis elevator (Fig 1).

The initial model [16,17] was tested on a human cadaver, revealing several limitations, including dimensions that made entry into the oral and laryngeal cavities difficult, the use of a silicone material that was not compatible with human tissue and/or mucosa, and high-hardness surfaces that generated friction both upon contact with tissues and during the passage of the orotracheal tube. Additional concerns were related to safety when considering the use of the device in a human model. Initial cadaveric testing was performed on two human cadavers (one male, one female) with heights between 150–180 cm and weights between 50–80 kg. Silicone was deemed incompatible because, although it performed adequately on mannequins, its higher hardness and surface characteristics could generate excessive friction against mucosa in human tissue, hindering device advancement through the oral cavity; in addition, silicone particulates could shed and lodge in the airway. Subject-matter experts and clinical advisors raised these safety concerns—specifically, the risk of mucosal injury and fragment detachment—which prompted a comprehensive redesign of the prototype.

Given the limitations identified in the initial test with a human cadaver, a comprehensive redesign of the device and its development process was necessary. This marked the beginning of an iterative approach grounded in interdisciplinarity, which enabled refinement of both the technical and ergonomic aspects of the device. From that point onward, the project advanced under a structured methodology guided by clinical, biomechanical, and technical feasibility criteria to achieve a functional and safe solution for real-world application. This interdisciplinary effort involved anesthesiologists with expertise in airway management, biomedical engineers, and an industrial designer (see author affiliations).

The research team, composed of expert anesthesiologists specializing in airway management, biomedical engineers, and industrial designers, defined the operational variables to be evaluated during each iteration. Device modeling was carried out using *SolidWorks* CAD software (version 2022, Dassault Systèmes, Vélizy-Villacoublay, France), optimizing the geometry based on anatomical and functional parameters. In each design iteration, additive manufacturing techniques

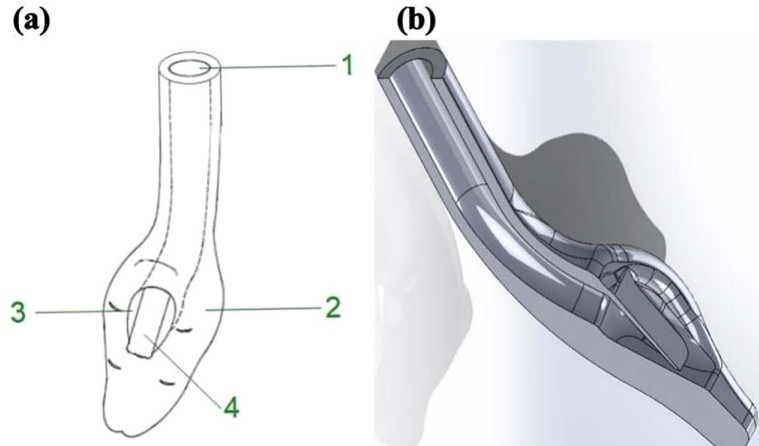

**Fig 1. Orolaryngeal ramp device schematics (not to scale). (a)** Main components: 1—External opening for the endotracheal tube; 2—Cup (contact surface with oropharyngeal tissues); 3—Laryngeal aperture (internal opening aligned with the glottic inlet); 4—Epiglottis elevator (semi-rigid flap). **(b)** CAD model illustrating the ramp angle and cup curvature optimized across iterations. The schematic represents the device geometry; specific printed materials used in later iterations (TPU 95A, BioMed Elastic 50A, BioMed Flex 80A) are detailed in the text and Table 2. On-figure annotations label components and geometric parameters.

were applied through rapid prototyping using a *Form 4B* 3D printer (Formlabs Inc., Somerville, MA, USA), allowing for the evaluation of ergonomic and mechanical aspects at each stage of development. Material selection was informed by manufacturer technical data sheets and biocompatibility documentation for BioMed Elastic 50A and BioMed Flex 80A resins (ISO 10993, USP Class VI, produced in an FDA-registered, ISO 13485 facility). Validation tests for each iteration were conducted using the *Laerdal Deluxe Difficult Airway Trainer* mannequin (Laerdal® Medical AS, Stavanger, Norway), as well as anatomical specimens from human cadavers. All cadaveric testing and validation procedures were conducted between February and June 2024 in the morphology laboratory, under protocols approved by the Comité de Ética, Bioética e Investigación of the Hospital Universitario Hernando Moncaleano Perdomo (Act No. 06-02, June 22, 2023).

In each validation, the following aspects were assessed: the intubation technique, ease of device insertion, ease of orotracheal tube passage in terms of friction, and the final anatomical position of the internal opening. Anatomical verification of the device's position was performed through direct inspection with a laryngoscope and endoluminal evaluation using a 0.24-inch [6.10 mm] diameter articulated borescope (NIDAGE). A 5-point Likert scale (anchors: 'very difficult' to 'very easy') was used to rate (i) ease of device insertion, (ii) perceived friction during tube passage, and (iii) overall ease-of-use [22]. Each iteration was assessed by five authors: Author 1 (clinical anesthesiology/airway management), Authors 2 and 3 (clinical sciences/public health research), Author 4 (industrial design), and Author 5 (biomedical/electronic engineering). Evaluations were performed independently and recorded immediately after each bench or cadaver test; when judgments differed, the majority opinion prevailed. By domain, Author 1 examined clinical feasibility, mucosal safety, and alignment with the glottic inlet; Authors 2–3 reviewed protocol adherence, reproducibility, and clarity of operational definitions; Author 4 evaluated ergonomics, insertion trajectory, and the cup–ramp interface; and Author 5 reviewed geometric tolerances, printability, and material response. All five authors participated in the final consensus for each iteration.

Finally, a high-performance prototype was developed to validate the functionality of the orolaryngeal ramp device for blind orotracheal intubation in human cadaver models, ensuring compliance with the required safety and efficacy standards.

## Ethics statement

This study was reviewed and approved by the Comité de Ética, Bioética e Investigación of the Hospital Universitario Hernando Moncaleano Perdomo, in accordance with the ethical regulations of the Colombian Ministry of Health (Resolutions 8430 of 1993 and 2378 of 2008) and the UNESCO Universal Declaration on Bioethics and Human Rights (Act No. 06-02, June 22, 2023). The study involved validation of a medical device using human cadaveric specimens and airway simulators, without recruitment of living human participants; informed consent procedures were not required. All cadaveric testing and validation procedures were conducted between February and June 2024 in the morphology laboratory, under approved institutional protocols.

## Results

The evaluation attributes considered in each iteration were defined by the research team based on previously established functional criteria (Table 1). Among the characteristics analyzed were the overall dimensions of the device, the presence or absence of key structural elements such as the epiglottis elevator and the air bubble chamber, as well as the mechanical behavior of the prototype during interaction with tissues and the orotracheal tube. Both qualitative and quantitative assessments were conducted to evaluate the degree of friction during insertion, positional stability, and ease of device manipulation. Additionally, the performance of each model was observed in relation to the intubation technique used, considering parameters such as the angle of entry and intubation success. These variables informed the iterative redesign process, enabling progressive improvements in the device's functionality and safety.

Ten iterations were required during the technological development process (Table 2). In the first four iterations, the fused deposition modeling technique was used with TPU 95A (Thermoplastic Polyurethane) materials. In the following

**Table 1. Operative variables used for evaluating the iterations.**

|  | Dimension | Epiglottis elevator | Air bubble | Friction with the tube | Friction with the tissue | Efficiency | Intubation technique | Learning Curve |
|---|---|---|---|---|---|---|---|---|
| **Iteration 1** | X |  |  | High |  | No | X |  |
| **Iteration 2** | X |  |  | High |  |  |  |  |
| **Iteration 3** | X | X |  | High |  | No |  |  |
| **Iteration 4a** | X | X |  | High |  | No |  |  |
| **Iteration 4b** | X | X | X | Medium | Medium | No | X |  |
| **Iteration 5** | X | X | X | Medium | Medium | +/- | X |  |
| **Iteration 6** | X |  | X | Medium | Medium | +++ | X |  |
| **Iteration 7** | X | X | X | Medium | Medium | +/- | X |  |
| **Iteration 8** | X |  | X | Low | Low | + | X |  |
| **Iteration 9** | X |  | X | Low | Low | +++ | X |  |
| **Iteration 10 (Final)** | X |  | X | Low | Low | +++ | X | X |

*X: Feature evaluated in the iteration.

+/-: Inconclusive tests.

+: Intubation success in more than 50% of cases.

+++: Intubation success in more than 95% of cases.

Air bubble refers to a sealed internal hollow feature within the cup intended to modulate wall compliance during insertion. Friction denotes a qualitative ordinal rating of perceived resistance during device insertion and tube passage [low/medium/high or Likert-based]. Assessments were performed by Autors 1,3,5.

**Table 2. Prototype iterations summary.**

|  | Cup Angle | Ramp angle | Epiglottis elevator | Material | Simulator | Cadaver | Observation |
|---|---|---|---|---|---|---|---|
| **Iteration 1** | 22.68° | 52.68° | With | TPU | OK | Failed | Could not enter the oral cavity. |
| **Iteration 2** | 22.68° | 52.68° | With | TPU | OK | Failed | Variation in the tip of the cup. |
| **Iteration 3** | 37.68° | 67.68° | With | TPU | OK | Failed | Uniform 3.5 mm perimeter reduction of the cup. |
| **Iteration 4a** | 22.68° | 60° | With | TPU | OK | Failed | Cup wall thickness adjusted to 2.5 mm. |
| **Iteration 4b** | 37.68° | 67.68° | With | TPU | OK | Failed | Adjustment of the posterior wall of the cup. |
| **Iteration 5** | 37.68° | 70° | With | Elastic 50A | OK | Failed | Ramp redesign, widening of the angled neck radius. |
| **Iteration 6** | 37.68° | 70° | Without | Elastic 50A | OK | Failed | No epiglottis elevator; adjustment of the posterior face of the cup. |
| **Iteration 7** | 47.67° | 80° | With | Elastic 50A | OK | Failed | The ramp position moved forward within the cup. |
| **Iteration 8** | 47.67° | 80° | Without | Elastic 80A | OK | OK | Cup redesign. Initial cadaveric intubation validations. |
| **Iteration 9** | 47.67° | 80° | With/ Without | Elastic 80A | OK | OK | Ramp moved forward. Variation in cup design, validation of three epiglottis elevator lengths. |
| **Iteration 10 (Final)** | 47.67° | 80° | With/ Without | Elastic 80A | OK | OK | Adjustment of the tip and alignment indicator. |

TPU: Thermoplastic Polyurethane. "Simulator OK" = successful intubation in airway simulator; "Cadaver OK" = successful intubation in human cadaver model.

three iterations, BioMed Elastic 50A Resin was used, which was manufactured in facilities registered with the U.S. Food and Drug Administration (FDA). Finally, in the last three iterations, BioMed Flex 80A Resin was used, a firm and flexible medical-grade material certified under ISO 10993, suitable for long-term skin contact (over 30 days) and short-term contact with mucous membranes (less than 24 hours).

## Device iterations using TPU 95A

Based on the characteristics and limitations of the device observed in the human cadaver, along with an analysis of technical factors such as dimensions and material type of the utility model, TPU 95A material was used during the first four iterations. This material is a semi-flexible, chemically resistant filament with strong layer adhesion, commonly used in additive manufacturing for biomedical applications [23].

At this stage, significant difficulties were observed in most of the evaluated variables, including resistance to the passage of the endotracheal tube due to a smaller radius of the internal channel arc and a reduced angle of the internal opening, as well as the rigidity of the material, which caused damage to the tube's cuff. Additionally, challenges were encountered during device insertion, such as high friction against the tongue, increased force required to advance the device, resulting in mechanical airway obstruction; the need for additional maneuvers to open the mouth; the use of neutral or hyperextended neck positioning techniques; and the need to manually displace the tongue.

To improve the device's functionality, these iterations increased the internal inclination angle of the orolaryngeal ramp to 30 degrees relative to the cup, as well as the radius of the laryngeal opening and the internal channel for the passage of the orotracheal tube. Additionally, the distal end of the device was adjusted by slightly widening it to better conform to the anatomy of the lower larynx.

As a result of this phase, the external volume of the cup was reduced, particularly in its dorsal portion, thereby creating an internal angle at the laryngeal opening with a wider tube channel radius. This adjustment aimed to facilitate device insertion and ensure the effective passage of the orotracheal tube without causing damage. Fig 2a shows the progression of prototypes manufactured during the iterations using TPU (iterations 1 to 4b). Fig 2b shows one of the tests performed on the mannequin.

## Device iterations using BioMed Elastic 50A

During iterations 5, 6, and 7, the material was changed to BioMed Elastic 50A Resin, a biocompatible material with properties compatible with human tissue and environmentally friendly characteristics. In this phase of the process, an internal air bubble was added to the posterior wall of the device's cup as a key feature to reduce difficulty during passage between the teeth and the tonsillar pillars. After completing the iterations, the research team decided to retain the air bubble feature because mannequin testing suggested easier device insertion between the teeth and tonsillar pillars and smoother tube advancement. Subsequent iterations focused on modifying the cup, the epiglottis elevator, and the ramp angle. However,

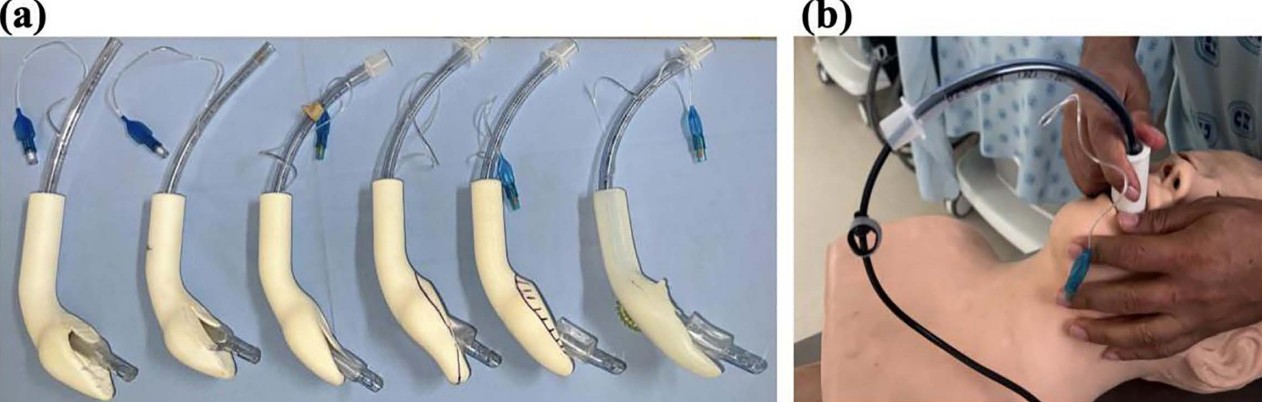

**Fig 2. Designs with Thermoplastic Polyurethane (TPU 95A). (a)** Iterations 1–4b (left to right), showing progressive changes in orotracheal tube exit angle (α) and cup geometry; labels 1, 2, 3, 4a, 4b are shown on-figure. **(b)** Example test on the mannequin.

modifications were made to the cup, the epiglottis elevator, and the ramp angle. Nonetheless, mannequin testing revealed that the material used exhibited structural damage in the device.

For iteration number 7, the test results based on the Likert scale [22], as evaluated by the research team, indicated that most participants had a positive perception and a good experience regarding the ease of inserting the device into the oral cavity and passing the orotracheal tube through it. Additionally, it was found that the insertion technique required a neutral head position, with minimal manipulation of the mouth opening and the use of gel lubricant both internally and externally. The air bubble allowed the device to conform to the posterior pharyngeal wall, helping to prevent injuries. The tube exit angle was appropriate and enabled successful orotracheal intubation in 100% of tested cases, with proper alignment toward the glottic opening (Fig 3).

### Device iterations using BioMed Flex 80A

Considering the results from iteration 7 and the presence of tears and relative fragility in the device associated with the material, it was decided to use BioMed Flex 80A Resin starting from iteration 8. This is a medical-grade material that is flexible, durable, tear-resistant, and capable of withstanding temperatures up to 100 degrees Celsius. It has a low likelihood of causing allergic reactions and requires disinfection with 70% isopropyl alcohol for 5 minutes.

Iterations 9 and 10 introduced modifications to the upper edge of the internal opening, the tube exit ramp, and the cup with an air bubble to ensure proper alignment with the laryngeal walls. However, it was observed that the tip of the prototype caused some abrasive injuries to the tissue of the cadaveric anatomical model. As a result, the tip of the device was modified and rounded (Fig 3).

The final iteration passed all tests regarding ease of insertion into the oral cavity, ease of passage of the orotracheal tube through the device, and achieved successful orotracheal intubation in 100% of cases performed on both airway simulators and cadaveric models. It also showed reduced friction (using gel- and water-based lubricants), a sharp ramp angle, and proper alignment of the internal opening with the glottic aperture.

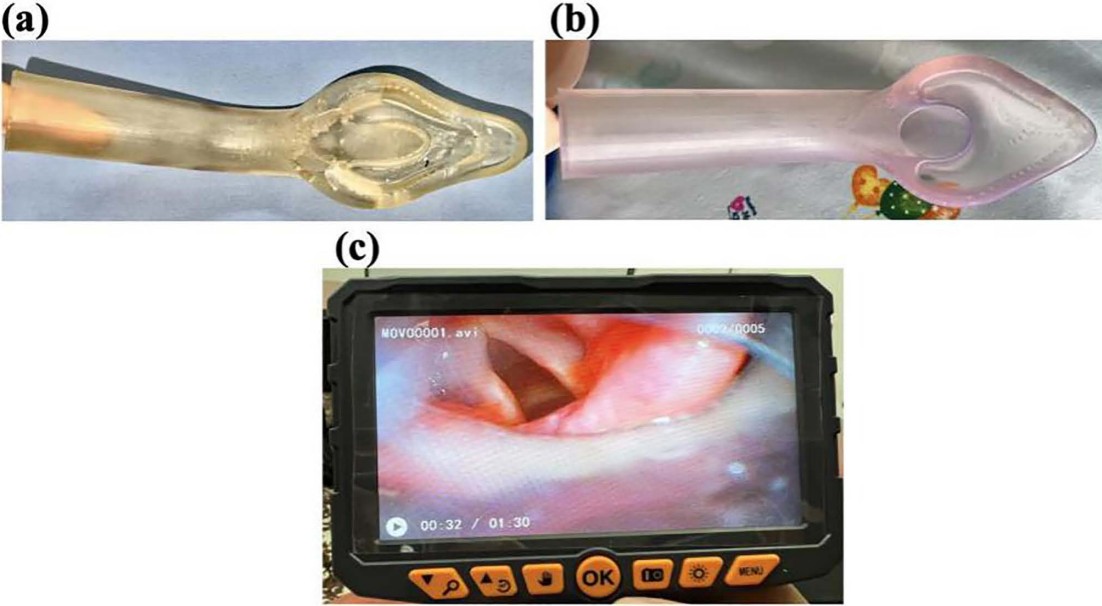

**Fig 3. Designs with BioMed Elastic Resin.** (a) 50A; (b) 80A; (c) Flexible bronchoscopy image through the prototype in a human cadaver.

 7 / 12

Based on this evaluation process, the final 3D-printed prototype design was selected, featuring a 47.67° cup angle, an 80° ramp angle, and adjustments to both the tip and the alignment indicator. This final design was subsequently tested on anatomical specimens from human cadavers (Fig 4).

## Discussion

The generation of new scientific knowledge in the medical field is closely linked to the ability to develop and integrate emerging technologies into complex clinical settings. However, the skills and expertise acquired by airway operators remain essential for safe and effective care, particularly in emergency scenarios [24]. In the airway, although the use of indirect visualization devices such as videolaryngoscopes and fiberoptic bronchoscopes has been shown to improve endotracheal intubation success rates, it can also limit the operator's cognitive autonomy in the event of technological failures or adverse clinical conditions [19].

On the other hand, SGDs have been relegated to rescue roles within airway management algorithms, primarily due to their association with a higher risk of pulmonary aspiration [10]. Blind intubation maneuvers, whether manual or performed through SGDs, have been progressively abandoned in favor of direct visualization of the tube passing between the vocal cords, which is considered the most reliable sign for preventing esophageal intubation [20]. Nevertheless, even with the use of direct or indirect visualization devices, limitations may still arise, such as incomplete glottic view, secretions, blood, or unfavorable anatomical configurations that hinder successful intubation [20].

A recent systematic review [15] showed that the effectiveness of SGDs for blind intubation varies depending on the type of device, the operator's profile, and the patient population involved. It highlighted 100% success rates in devices such as the Fastrach and I-Gel, with average intubation times under 30 seconds in some studies.

These findings are particularly relevant considering that, in emergencies, time is a critical factor: it is estimated that a well-executed videolaryngoscopy takes an average of 38 seconds, and a fiberoptic intubation is considered efficient if completed in under 120 seconds [19]. In our study, procedural time was not measured; the <30 s target was an a priori design rationale informed by prior literature rather than an outcome assessed here. However, incorrect positioning of SGDs is a common cause of failure in blind intubation, although they still allow for effective ventilation. This situation can be explained by factors such as a folded epiglottis inside the laryngeal mask or its adherence to the posterior pharyngeal wall, as demonstrated by magnetic resonance imaging in sedated or anesthetized patients [25].

In this context, the orolaryngeal ramp device described in this work was conceived as an innovative technological solution through an iterative development strategy [16,17]. The methodology employed integrated CAD techniques for three-dimensional modeling, enabling rapid and precise iterations on both the internal and external geometry of the

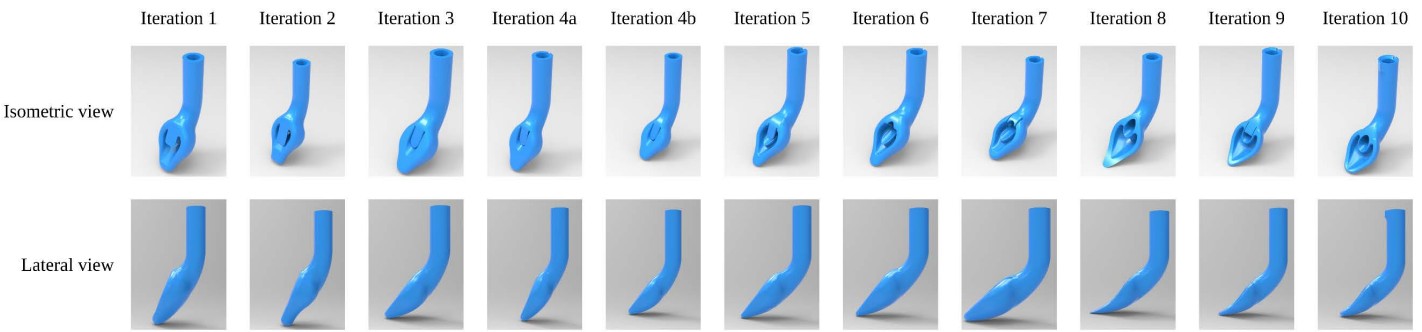

**Fig 4. The ten different designs.** CAD models and renders.

device, optimizing its structure based on the expected behavior within the upper airway. Subsequently, rapid prototyping methods using 3D printing were applied for the physical fabrication of the device, which facilitated the evaluation of multiple versions over short periods and enabled functional validation in simulation models and human cadaveric specimens. This strategy aligns with scientific literature that recognizes AM as a transformative tool in the design of medical devices, due to its ability to accelerate development cycles, reduce initial manufacturing costs, and allow for personalized adaptations according to patient anatomy and clinical needs [26–28].

The orolaryngeal ramp was, a priori, designed to facilitate blind intubation in under 30 seconds (design target), minimize apnea time, and reduce the risk of aspiration. Its semi-rigid structure allows for efficient insertion by aligning the cup with the midline of the pharynx and lifting the epiglottis without requiring excessive cervical movement, an essential feature in patients with trauma or suspected spinal cord injury. Additionally, it facilitates the passage of the orotracheal tube, although internal and external lubrication are required for optimal performance. However, the methodological limitations of the study must be acknowledged, particularly the qualitative nature of some evaluated variables, which led to divergent interpretations among researchers, as well as the specificity of the criteria applied to each device. These criteria reflect the problem-solution approach typical of applied medical engineering.

From a sustainability perspective, the device was developed using biocompatible and sterilizable materials, with potential for multiple uses, an advantage over most currently available SGDs, especially in resource-limited settings [26]. This approach addresses not only efficacy criteria but also ethical and economic considerations related to bio-cost-effectiveness and the responsible management of healthcare resources [27].

The orolaryngeal ramp's role as an educational tool is particularly noteworthy in airway management training scenarios. Its development aims to facilitate intubation in emergencies by non-expert operators, which is especially relevant given the variability in professional training and the limited availability of technological resources. Its ergonomic and intuitive design, coupled with a short learning curve, enables the operator to recognize complementary signs of successful tracheal intubation even in the absence of direct vocal cord visualization, while simultaneously minimizing the risk of cervical injury by avoiding aggressive neck extension maneuvers [24].

Based on the results of this study, it is recommended that clinical validation studies be conducted for the device, assessing safety aspects in both hospital and prehospital settings, and more precisely characterizing its technical and operational limitations across different anatomical profiles and operator expertise levels. These steps will be essential to determine its integration into airway management algorithms and its scalability as a sustainable and effective medical technology in emergency scenarios. Prototype assessments during the development phase were performed by members of the development team, which may introduce observer bias. Future work will prioritize usability and performance evaluations conducted by independent end users (e.g., first responders and non-developer clinicians) to minimize bias and improve external validity.

## Potential applications

Beyond the present preclinical validation, this device may serve (i) as a low-cost adjunct for simulation-based training and skills maintenance in blind intubation; (ii) as a field or prehospital option in austere and resource-limited environments where visualization tools are unavailable or impractical; and (iii) as a size-scalable platform that could be adapted to different patient populations, including pediatric airways. These potential uses will require dedicated studies of usability, safety, sterilization/reuse protocols, and age-specific anatomical fit—ideally with independent end users—before any clinical adoption.

## Emergency relevance

The intended clinical context for this device is emergency and prehospital airway management when visualization tools (e.g., video- or fiber-optic laryngoscopy) are unavailable, impractical, or delayed. By guiding the endotracheal tube along

an extraglottic ramp with minimal cervical manipulation, the approach may offer a rapid, low-resource adjunct for blind orotracheal intubation. Confirmatory studies with independent end users in realistic scenarios are needed to define its role relative to existing options.

## Conclusions

This study offers an initial, transparent account of the device's design and cadaveric validation. Its value will be strengthened by independent end-user testing and fuller methodological reporting. Our iterative development of the orolaryngeal ramp illustrates a practical pathway—combining CAD-guided design and rapid 3D printing—to prototype and refine an extraglottic aid for blind orotracheal intubation in emergency airway management. Its validation in simulation models and human cadaveric specimens provides preliminary evidence of its applicability, resulting in an innovative, functional, and cost-effective solution, particularly in contexts where access to technology is limited or compromised.

Furthermore, its ergonomic design and reusability make it a valuable tool for clinical training and medical education, offering a viable and sustainable alternative for strengthening competencies in airway management, especially in low- and middle-complexity settings. While the device demonstrated functional feasibility in simulators and cadaveric models, real-world clinical evaluation should proceed only after addressing the methodological gaps identified here (greater procedural detail, independent end-user testing, and comprehensive reporting).

By coupling CAD-guided iteration with biocompatible 3D printing, this work outlines a reproducible pathway to rapidly prototype and refine extraglottic airway adjuncts. If future studies confirm safety, usability, and performance with independent end users and in real-world conditions, the potential clinical impact includes expanded training capacity, additional prehospital options when visualization is not feasible, and format variants tailored to different patient groups, including pediatric applications. Pending validation in real-world emergency and prehospital settings, this platform could expand airway options when visualization is limited, complementing established techniques rather than replacing them.

## Acknowledgments

The authors thank Professor Alba Rocio Valencia, coordinator of the morphology laboratory at Universidad Surcolombiana, for her assistance with the cadaveric models used in the testing.

## Author contributions

**Conceptualization:** Jorman H. Tejada-Perdomo, Valentina Gutierrez-Perdomo, Jorge A. Pérez-Gamboa, Jorge A. Ramos-Castaneda.

**Data curation:** Jorman H. Tejada-Perdomo, Valentina Gutierrez-Perdomo, Juana V. Agudelo-Castro, Jorge A. Ramos-Castaneda.

**Formal analysis:** Jorman H. Tejada-Perdomo, Valentina Gutierrez-Perdomo, Alejandro Weinstein, Jorge A. Ramos-Castaneda.

**Funding acquisition:** Jorman H. Tejada-Perdomo, Alejandro Weinstein.

**Methodology:** Jorman H. Tejada-Perdomo, Valentina Gutierrez-Perdomo, Juana V. Agudelo-Castro, Jorge A. Pérez-Gamboa, Jorge A. Ramos-Castaneda.

**Project administration:** Jorman H. Tejada-Perdomo, Valentina Gutierrez-Perdomo.

**Resources:** Jorman H. Tejada-Perdomo, Valentina Gutierrez-Perdomo, Jorge A. Pérez-Gamboa.

**Supervision:** Jorman H. Tejada-Perdomo, Valentina Gutierrez-Perdomo, Alejandro Weinstein, Jorge A. Ramos-Castaneda.

**Visualization:** Jorman H. Tejada-Perdomo, Valentina Gutierrez-Perdomo, Jorge A. Pérez-Gamboa.

**Writing – original draft:** Jorman H. Tejada-Perdomo, Valentina Gutierrez-Perdomo, Alejandro Weinstein, Jorge A. Ramos-Castaneda.

**Writing – review & editing:** Jorman H. Tejada-Perdomo, Alejandro Weinstein, Sebastián San Martín, Rodrigo Salas, Jorge A. Ramos-Castaneda.

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
