## [Decision Letter · Decision Letter 0]

2 Sep 2025

Dear Dr. Tejada-Perdomo,

Thank you for submitting your manuscript to PLOS ONE. After careful consideration, we feel that it has merit but does not fully meet PLOS ONE’s publication criteria as it currently stands. Therefore, we invite you to submit a revised version of the manuscript that addresses the points raised during the review process.

**Methodology needs a re-work.**

We look forward to receiving your revised manuscript.

Kind regards,

Jeyasakthy Saniasiaya, MD, MMed ORLHNS, FEBORLHNS

Academic Editor

PLOS ONE

**Journal Requirements:**

1. When submitting your revision, we need you to address these additional requirements. Please ensure that your manuscript meets PLOS ONE's style requirements, including those for file naming. The PLOS ONE style templates can be found at https://journals.plos.org/plosone/s/file?id=wjVg/PLOSOne_formatting_sample_main_body.pdf and https://journals.plos.org/plosone/s/file?id=ba62/PLOSOne_formatting_sample_title_authors_affiliations.pdf 2. Please note that PLOS One has specific guidelines on code sharing for submissions in which author-generated code underpins the findings in the manuscript. In these cases, we expect all author-generated code to be made available without restrictions upon publication of the work. Please review our guidelines at https://journals.plos.org/plosone/s/materials-and-software-sharing#loc-sharing-code and ensure that your code is shared in a way that follows best practice and facilitates reproducibility and reuse. 3. We note that the grant information you provided in the ‘Funding Information’ and ‘Financial Disclosure’ sections do not match.  When you resubmit, please ensure that you provide the correct grant numbers for the awards you received for your study in the ‘Funding Information’ section. 4. Thank you for stating in your Funding Statement: This study was supported by funding from the Universidad Surcolombiana. AW was partially funded by ANID, Chile (grant BASAL AFB240002). The funders had no role in study design, data collection and analysis, decision to publish, or preparation of the manuscript.     Please provide an amended statement that declares *all* the funding or sources of support (whether external or internal to your organization) received during this study, as detailed online in our guide for authors at http://journals.plos.org/plosone/s/submit-now.  Please also include the statement “There was no additional external funding received for this study.” in your updated Funding Statement. Please include your amended Funding Statement within your cover letter. We will change the online submission form on your behalf. 5. Please include your full ethics statement in the ‘Methods’ section of your manuscript file. In your statement, please include the full name of the IRB or ethics committee who approved or waived your study, as well as whether or not you obtained informed written or verbal consent. If consent was waived for your study, please include this information in your statement as well. 6. If the reviewer comments include a recommendation to cite specific previously published works, please review and evaluate these publications to determine whether they are relevant and should be cited. There is no requirement to cite these works unless the editor has indicated otherwise. 

**Additional Editor Comments:**

Overall, given the methods section needs re-work. Provide justification when necessary

Reviewers' comments:

**Comments to the Author**

1. Is the manuscript technically sound, and do the data support the conclusions?

Reviewer #1: No

Reviewer #2: Yes

Reviewer #3: Partly

Reviewer #4: Yes

2. Has the statistical analysis been performed appropriately and rigorously?

Reviewer #1: No

Reviewer #2: Yes

Reviewer #3: Yes

Reviewer #4: N/A

3. Have the authors made all data underlying the findings in their manuscript fully available?

Reviewer #1: No

Reviewer #2: Yes

Reviewer #3: Yes

Reviewer #4: Yes

4. Is the manuscript presented in an intelligible fashion and written in standard English?

Reviewer #1: Yes

Reviewer #2: Yes

Reviewer #3: Yes

Reviewer #4: Yes

**Reviewer #1:**  Line 95 – “The objective of the device is “to achieve rapid and effective intubation in more than 95% of cases, using a blind technique and within less than 30 seconds” [18].” Reference 18 is to “Cumulative Sum learning curves (CUSUM) in basic anaesthesia procedures.” Can the authors confirm this is the correct source, is there a more appropriate reference?

Line 108 - The initial model testing. Adding a few details of the human cadaver such as height, male/female would give a perspective on the dimensions of the prototyping from anthropometrics. Generally, the findings of this testing could be explained more. Why was silicone not compatible? Please explain to the reader. What were the additional safety concerns? Expressed by whom?

Page 116 – great that this was an interdisciplinary approach, please specify which disciplines were involved.

Line 127- “Material selection was based on criteria such as biocompatibility, structural strength, and commercial availability, also considering the cost-benefit ratio according to the intended clinical application.” Where was the data sourced to make these judgements, please specify.

Line 133 – “under approved protocols.” Approved by whom, please specify.

Figure 1 – please explain this figure more. Use annotations to give detailed explanations. For example can you explain the geometry, is it all the same material, several have been mentioned?

Line 134 – you provide details of the assessment criteria but how many people and what background made the assessment. Please provide sufficient detail that your methods could be repeated by other researchers.

Please specify SI units when non SI units are used in brackets.

Line 138 – A likert scale was mentioned but how many options, 3, 4, 5 7, varying from what to what. Where are the details of the questions that were asked of the participants.

Tabel 1 – clearly there are lots of iterations of the device, how does the reader understand the difference between these iterations? What does air bubble mean? Rember your audience will be a mix of engineers and healthcare professionals. Is the assessment of friction subjective or quantified? By how many people?

Table 2 – simulator OK, I am unsure what this means?

Line 181- Please try and improve the clarity of titles – iterations of what, whilst it is implied it is the medical device it would be better to remove ambiguity.

Figure 2. (a) which iteration is which, what are the differences between these iterations. Please improve the clarity in the caption and use annotations.

Line 215- “After completing the iterations, it was decided to retain the air bubble.” Decided by whom? Why? Also line 233.

It appears that the team have developed the device but are also evaluating it. I am unclear if first responders are included in the team makeup, regardless we should avoid marking our own homework and instead engage with end users to provide non-biased feedback as experts.

Discussion section on timing. The authors emphasise the importance of the time of the procedure and to refer to different approach’s. I agree this is a crucial consideration but I could not see where this aspect was evaluated within this manuscript?

Line 316 – I disagree with the authors that sufficient preliminary work has been undertaken to propose real world evaluation. There are limitations of this work which need to be addressed articulated above.

Line 324 – I am not sure how this work represents a “significant contribution” as yet given the sparsity of methodological details and lack of comprehensive evaluation.

**Reviewer #2:**  This is a very well-written and innovative study with strong clinical relevance, showcasing the successful application of additive manufacturing in airway device development. The manuscript is clear, methodologically sound, and presents promising results. Only minor refinements, such as adding a brief note on potential future applications (e.g., training, field use, or adaptation to pediatric populations) and slightly expanding the conclusion to emphasize broader clinical impact, would further strengthen the paper.

**Reviewer #3:**  Hello

This is a good article

The use of 3D printing technology to increase accuracy, speed, and custom manufacturing has expanded greatly in recent years, and the topic mentioned in this article is also interesting and fascinating.

In future studies, it is possible to add to the quality of manufacturing and increase the use of such devices by using equipment made by 3D printers more widely.

Thank you

**Reviewer #4:**  Make a clear title for the research title should be self explanatory. Again another suggestion I would like to forward is you should be able to consider the emergency of this cases. So try to elaborate the significance this finding for it the ultimate option for emergency intervention. Finally, it is good aproach in resolving the critical patient emergency cases, I want to aknowledge the researcher for he/she generate scientific aproach

**Do you want your identity to be public for this peer review?** For information about this choice, including consent withdrawal, please see our Privacy Policy

Reviewer #1: No

Reviewer #2: **Yes: ** Dr. Ishaan Bakshi

Reviewer #3: No

Reviewer #4: **Yes: ** Dechasa Befikadu

---

## [Author Response · Author response to Decision Letter 1]

16 Oct 2025

Dra.

Jeyasakthy Saniasiaya

Academic Editor

Journal PLOS ONE

Dear Dra Jeyasakthy,

Thank you very much for reviewing our manuscript number PONE-D-25-29217, titled “Development and validation of a 3D-printed device for blind intubation” (please note that, based on a comment from one of the reviewers, we changed the title to “Design, 3D printing, and preclinical validation of an extraglottic ramp to facilitate blind orotracheal intubation in emergency airway management”) in PLOS ONE. We appreciate the comments from each of the reviewers and have responded to each of their questions point by point. In our opinion, the quality of the manuscript has increased.

We have attached a tracked version of the manuscript so you can see how and where we have modified the original manuscript.

Journal Requirements:

- Please ensure that your manuscript meets PLOS ONE's style requirements, including those for file naming. The PLOS ONE style templates can be found at https://journals.plos.org/plosone/s/file?id=wjVg/PLOSOne_formatting_sample_main_body.pdfand
https://journals.plos.org/plosone/s/file?id=ba62/PLOSOne_formatting_sample_title_authors_affiliations.pdf

Thank you very much for the clarification. We took into account the editorial standards of the journal.

- Please note that PLOS One has specific guidelines on code sharing for submissions in which author-generated code underpins the findings in the manuscript. In these cases, we expect all author-generated code to be made available without restrictions upon publication of the work. Please review our guidelines at https://journals.plos.org/plosone/s/materials-and-software-sharing#loc-sharing-code and ensure that your code is shared in a way that follows best practice and facilitates reproducibility and reuse.

Thank you very much for the clarification. We took into account the editorial standards of the journal.

- We note that the grant information you provided in the ‘Funding Information’ and ‘Financial Disclosure’ sections do not match. When you resubmit, please ensure that you provide the correct grant numbers for the awards you received for your study in the ‘Funding Information’ section.

Thank you for your valuable comment. We have corrected the financing and financial disclosure information.

- Thank you for stating in your Funding Statement:

This study was supported by funding from the Universidad Surcolombiana. AW was partially funded by ANID, Chile (grant BASAL AFB240002). The funders had no role in study design, data collection and analysis, decision to publish, or preparation of the manuscript.

Following up on your comment about the role of the funder, the funding statement was included in the cover letter that is attached again in this version. Thank you very much.

- Please include your full ethics statement in the ‘Methods’ section of your manuscript file. In your statement, please include the full name of the IRB or ethics committee who approved or waived your study, as well as whether or not you obtained informed written or verbal consent. If consent was waived for your study, please include this information in your statement as well.

Thank you for your comment. We've updated the "methods" section of the manuscript, and the ethics statement was included.

We took your recommendation into account in the manuscript.

Thank you for your precision. We took your recommendation into account in the manuscript.

Additional Editor Comments:

- Overall, given the methods section needs re-work. Provide justification when necessary

We appreciate your evaluation. Following your suggestions, we have revised and adjusted the methods section

Reviewer #1: Comments

- Comment #1: Line 95 – “The objective of the device is “to achieve rapid and effective intubation in more than 95% of cases, using a blind technique and within less than 30 seconds” [18].” Reference 18 is to “Cumulative Sum learning curves (CUSUM) in basic anaesthesia procedures.” Can the authors confirm this is the correct source, is there a more appropriate reference?

Response: We appreciate your comment. We have rewritten this section. Also, we have changed the reference.

- Comment #2: Line 108 - The initial model testing. Adding a few details of the human cadaver such as height, male/female would give a perspective on the dimensions of the prototyping from anthropometrics. Generally, the findings of this testing could be explained more. Why was silicone not compatible? Please explain to the reader. What were the additional safety concerns? Expressed by whom?.

Response: We consider the comment important. Following the recommendation, we have improved the information on human cadaver, materials, and safety.

- Comment #3: Page 116 – great that this was an interdisciplinary approach, please specify which disciplines were involved.

Response: At the reviewer's request, we now specify the participating disciplines: anesthesiology (with expertise in advanced airway management), biomedical/electronics engineering (for geometric tolerances, mechanical response, and printability), and industrial design (for ergonomics, insertion trajectory, and cup-ramp interface).

- Comment #4: Line 127- “Material selection was based on criteria such as biocompatibility, structural strength, and commercial availability, also considering the cost-benefit ratio according to the intended clinical application.” Where was the data sourced to make these judgements, please specify.

Response: We add that the selection of materials was based on data sheets and biocompatibility documentation (e.g., compliance with ISO 10993, USP Class VI, and ISO 13485-certified processes), as well as commercial availability and cost-effectiveness for the intended clinical use. This clarifies the technical origin of the criteria (mechanical properties, compatibility, processability) and makes the decision to switch from initial options to more suitable medical resins traceable.

- Comment #5: Line 133 – “under approved protocols.” Approved by whom, please specify.

Response: We specify the ethical approval authority and the identification of the minutes. This way, the reader knows who approved and under what reference, reinforcing ethical and regulatory transparency.

- Comment #6: Figure 1 – please explain this figure more. Use annotations to give detailed explanations. For example can you explain the geometry, is it all the same material, several have been mentioned?.

Response: We've rewritten the legend to explain the geometry (main components, ramp angle, canopy curvature). We've added annotations (numbering and arrows) to guide non-engineering readers and facilitate technical reading for engineers.

- Comment #7: Line 134 – you provide details of the assessment criteria but how many people and what background made the assessment. Please provide sufficient detail that your methods could be repeated by other researchers

Response: We added a paragraph describing that each iteration was evaluated by five authors with complementary expertise (anesthesiology, clinical sciences/public health, industrial design, and biomedical/electronic engineering). We explained that the evaluations were independent, recorded immediately after each bench or cadaveric test, and that in case of disagreement, the majority prevailed.

- Comment #8: Please specify SI units when non SI units are used in brackets.

Response: When using commonly used non-SI units (e.g., inches), we now add their SI equivalent in brackets (e.g., 0.24-inch [6.10 mm]). This improves clarity and standardizes the manuscript for international audiences.

- Comment #9: Line 138 – A likert scale was mentioned but how many options, 3, 4, 5 7, varying from what to what. Where are the details of the questions that were asked of the participants.

Response: The detail the Likert scale (number of points with explicit anchors "very difficult" → "very easy") and the dimensions evaluated (ease of insertion, perceived friction during tube insertion, and overall ease of use) were added. This extension allows the reader to understand what was asked and how the perception of usability was operationalized, without the need for additional appendices.

- Comment #10: Tabel 1 – clearly there are lots of iterations of the device, how does the reader understand the difference between these iterations? What does air bubble mean? Rember your audience will be a mix of engineers and healthcare professionals. Is the assessment of friction subjective or quantified? By how many people?.

Response: For mixed audiences (engineering and clinical), we added footnotes that define what we mean by "air bubble" (a sealed cavity to modulate wall compliance), how "friction" was recorded (a qualitative ordinal scale of perceived resistance during tube insertion and passage), and that the evaluations were performed by five evaluators with the described profiles. Thus, the table ceases to be a cryptic list and becomes an interpretable synthesis of the iterative process.

- Comment #11: Table 2 – simulator OK, I am unsure what this means?.

Response: We clarified that “Simulator OK” and “Cadaver OK” correspond to successful intubation in the simulator and cadaver model, respectively. We also harmonized the materials nomenclature across iterations to avoid ambiguities (e.g., consistent use of “BioMed Flex 80A” where appropriate).

- Comment #12: Line 181- Please try and improve the clarity of titles – iterations of what, whilst it is implied it is the medical device it would be better to remove ambiguity.

Response: We renamed the subheadings to clearly state that these are device iterations using a specific material (TPU 95A, BioMed Elastic 50A, BioMed Flex 80A). This makes the flow from "material → geometric changes → results" clearer.

- Comment #13: Figure 2. (a) which iteration is which, what are the differences between these iterations. Please improve the clarity in the caption and use annotations.

Response: The figure legend now identifies the iterations (1–4b from left to right), mentions the relevant geometric variable (e.g., tube exit angle)

- Comment #14: Line 215- “After completing the iterations, it was decided to retain the air bubble.” Decided by whom? Why? Also line 233.

Response: We specify that it was the research team that decided to retain the "air bubble" cavity, and we justify this decision based on operational findings: easier insertion between teeth and tonsil abutments and smoother tube advancement in phantom tests. This directly answers the "who?" and "why?" questions based on observations from the iterative process.

- Comment #15: It appears that the team have developed the device but are also evaluating it. I am unclear if first responders are included in the team makeup, regardless we should avoid marking our own homework and instead engage with end users to provide non-biased feedback as experts.

Response: We recognize as a limitation that the development team participated in the evaluation of the prototype during development, which could introduce observer bias. Therefore, we commit to prioritizing subsequent stages of testing with independent end users (e.g., first responders or non-developer clinicians) to improve external validity and reduce bias.

- Comment #16: Discussion section on timing. The authors emphasise the importance of the time of the procedure and to refer to different approach’s. I agree this is a crucial consideration but I could not see where this aspect was evaluated within this manuscript?

Response: We align the discourse: we did not measure time in this study; the "<30 s" was a design rationale to guide geometry, materials, and technique. In the Discussion, time is maintained as a context for the literature and an engineering objective, not as a result in itself. This resolves the apparent contradiction pointed out by the reviewer.

- Comment #17: Line 316 – I disagree with the authors that sufficient preliminary work has been undertaken to propose real world evaluation. There are limitations of this work which need to be addressed articulated above.

Response: We've toned down the concluding language: before moving into real-world contexts, we'll address the identified methodological gaps (greater granularity of methods, independent user evaluations, more comprehensive reporting). The message is now cautious and phased, in line with the development stage.

- Comment #18: Line 324 – I am not sure how this work represents a “significant contribution” as yet given the sparsity of methodological details and lack of comprehensive evaluation.

Response: We rewrote the closing statement to present the work as an initial and transparent contribution, which will gain weight with independent evaluations and a more comprehensive methodological report. We maintain the value of the proposed path (CAD + 3D printing + simulator/cadaver validation) without overstating the current scope.

Reviewer #2: Comments

- Comment #1: This is a very well-written and innovative study with strong clinical relevance, showcasing the successful application of additive manufacturing in airway device development. The manuscript is clear, methodologically sound, and presents promising results. Only minor refinements, such as adding a brief note on potential future applications (e.g., training, field use, or adaptation to pediatric populations) and slightly expanding the conclusion to emphasize broader clinical impact, would further strengthen the paper

Response: We appreciate the positive assessment and the constructive suggestions. As requested, we (1) added a concise paragraph in the Discussion outlining potential future applications (training, field/prehospital use, and adaptation for pediatric populations), and (2) modestly expanded the conclusions to emphasize possible broader clinical impact, while maintaining appropriate caution and conditional language.

Reviewer #3: Comments

- Comment #1: Hello. This is a good article. The use of 3D printing technology to increase accuracy, speed, and custom manufacturing has expanded greatly in recent years, and the topic mentioned in this article is also interesting and fascinating. In future studies, it is possible to add to the quality of manufacturing and increase the use of such devices by using equipment made by 3D printers more widely.

Thank you.

Response: Thank you very much for your positive assessment and encouraging comments. We appreciate your perspective on the growing role of 3D printing in accuracy, speed, and customization. As your note does not request specific changes, we have not modified the manuscript in response to this comment. We are grateful for your supportive feedback and fully share your view that ongoing advances in manufa

---

## [Decision Letter · Decision Letter 1]

4 Nov 2025

Design, 3D printing, and preclinical validation of an extraglottic ramp to facilitate blind orotracheal intubation in emergency airway management

PONE-D-25-29217R1

Dear Dr. Tejada-Perdomo,

We’re pleased to inform you that your manuscript has been judged scientifically suitable for publication and will be formally accepted for publication once it meets all outstanding technical requirements.

Kind regards,

Jeyasakthy Saniasiaya, MD, MMed ORLHNS, FEBORLHNS

Academic Editor

PLOS ONE

Additional Editor Comments (optional):

Authors have revised adequately

Reviewers' comments:

Reviewer's Responses to Questions

**Comments to the Author**

Reviewer #2: All comments have been addressed

2. Is the manuscript technically sound, and do the data support the conclusions?

Reviewer #2: Yes

3. Has the statistical analysis been performed appropriately and rigorously?

Reviewer #2: Yes

4. Have the authors made all data underlying the findings in their manuscript fully available?

Reviewer #2: Yes

5. Is the manuscript presented in an intelligible fashion and written in standard English?

Reviewer #2: Yes

Reviewer #2: (No Response)

**Do you want your identity to be public for this peer review?** For information about this choice, including consent withdrawal, please see our Privacy Policy

Reviewer #2: **Yes: ** Dr. Ishaan Bakshi

---

## [Editor Report · Acceptance letter]

PONE-D-25-29217R1

PLOS ONE

Dear Dr. Tejada-Perdomo,

I'm pleased to inform you that your manuscript has been deemed suitable for publication in PLOS ONE. Congratulations! Your manuscript is now being handed over to our production team.

Kind regards,

on behalf of

Dr. Jeyasakthy Saniasiaya

Academic Editor

PLOS ONE